# *Myroides* species, pathogenic spectrum and clinical microbiology sight in Mexican isolates

**Claudia Adriana Colín-Castro**[1☯], **Jossue Mizael Ortiz-Álvarez**[2☯], **Cindy Fabiola Hernández-Pérez**[3], **Melissa Hernández-Durán**[1], **María de Lourdes García-Hernández**[1], **María Guadalupe Martínez-Zavaleta**[1], **Noé Becerra-Lobato**[1], **Mercedes Isabel Cervantes-Hernández**[1], **Graciela Rosas-Alquicira**[1], **Guillermo Cerón-González**[1], **Braulio Josué Méndez-Sotelo**[4], **Rodolfo García-Contreras**[5], **Rafael Franco-Cendejas**[6]*, **Luis Esaú López-Jácome**[1,7]*

1 Infectious Diseases Division, Clinical Microbiology Laboratory, Instituto Nacional de Rehabilitación Luis Guillermo Ibarra Ibarra, Mexico City, Mexico, 2 Programa "Investigadoras e Investigadores por México", Consejo Nacional de Humanidades, Ciencias y Tecnologías (CONAHCYT), Mexico City, Mexico, 3 Centro Nacional de Referencia de Inocuidad y Bioseguridad Agroalimentaria, Servicio Nacional de Sanidad, Inocuidad y Calidad Agroalimentaria (SENASICA), Tecámac, Mexico State, Mexico, 4 Infectious Diseases Division, Instituto Nacional de Rehabilitación Luis Guillermo Ibarra Ibarra, Mexico City, Mexico, 5 Microbiology and Parasitology Department, Bacteriology Laboratory, Medicine Faculty, Universidad Nacional Autónoma de México, Mexico City, Mexico, 6 Biomedical Research Subdirection, Instituto Nacional de Rehabilitación Luis Guillermo Ibarra Ibarra, Mexico City, Mexico, 7 Biology Department, Chemistry Faculty, Universidad Nacional Autónoma de México, Mexico City, Mexico

☯ These authors contributed equally to this work.
* esaulopezjacome@quimica.unam.mx (LEL-J); raffcend@yahoo.com (RF-C)

**Data Availability Statement:** This Whole Genome Shotgun project has been deposited at DDBJ/ENA/GenBank (https://www.ncbi.nlm.nih.gov/bioproject/

## Abstract

### Introduction

*Myroides* is a bacterial genus of opportunistic bacteria responsible for diverse infections including in the skin and soft tissues, urinary tract, cardiovascular system, and bacteremia, although the incidence of its reported infections is low, it is increasing, likely due the use of better bacterial identification methods, but also perhaps due an increase in its prevalence. In addition, their pathogenic role is limited in terms of reporting their microbial physiology, so the present work provides information in this regard in addition to the information that is available in the international literature.

### Objective

To describe the microbiological and genetic characteristics of seven different *Myroides spp.* clinical strains and comment on their phylum, pathogenic and resistance characteristics.

### Methods

Seven *Myroides* spp., strains associated with infections were included from 1/January/2012 to 1/January/20 and identified by miniaturized biochemistry and MALDI-ToF. Susceptibility tests were performed according to CLSI recommendations by broth microdilution. Whole genome sequencing was performed for each strain and bioinformatics analysis were performed.

?term=PRJNA1094452) under the accession numbers JBBPHI000000000, JBBPHJ000000000, JBBPHK000000000, JBBPHL000000000, JBBPHM000000000, JBBPHN000000000 and JBBYHJ000000000. The version described in this paper are JBBPHI010000000, JBBPHJ010000000, JBBPHK010000000, JBBPHL010000000, JBBPHM010000000, JBBPHN010000000 and JBBYHJ010000000.

**Funding:** The author(s) received no specific funding for this work.

**Competing interests:** The authors have declared that no competing interests exist.

## Results

Strains were identified at genus level by two methodologies. Our results revealed that likely four strains belong to the species *Myroides odoratimimus*, while the other two may be undescribed ones. Remarkably, all isolates harbored several genes encoding antibiotic resistance determinants for ß-lactams, aminoglycosides and glycopeptides and in concordance, presented high levels of resistance, against these antibiotics (AK and GN both 100%, ATM, CAZ and FEP 100%, e.g.); moreover, the presences of carbapenemases were evidenced by meropenem (mCIM) and imipenem (CARBA NP) degrading activity in six isolates and two strains possessed plasmids harboring mainly ribosomal RNA genes, tRNAs and genes encoding proteins with unknown functions.

## Conclusions

Our study increases the knowledge about the biology of this understudied genus and highlights the potential of *Myroides* to emerge as a broader cause of recalcitrant opportunistic infections.

## Introduction

The genus *Myroides*, first isolated in 1923 in feces of patients with gastrointestinal infections [1], belongs to phylum *Bacteroidota*, class *Flavobacteriia*, order *Flavobacteriales* and family *Flavobacteriaceae*. *Myroides* is comprised of 13 species in NCBI Taxonomy Browser, which comprise *Myroides albus*, *Myroides fluvii*, *Myroides gitamensis*, *Myroides guanonis*, *Myroides indicus*, *Myroides injenensis*, *Myroides marinus*, *Myroides odoratimimus*, *Myroides odoratus*, *Myroides oncorhynchi*, *Myroides pelagicus*, *Myroides phaeus* and *Myroides profundi* [2, 3]. *Myroides* was initially included within *Flavobacterium* genus, specifically *Flavobacterium odoratum* and described by Stutzer in 1929, however, Vancanneyt and coworkers reclassified it and created a new genus, *Myroides* [4]. The given name came from Greek noun *mýron* which means perfume and Latin suffix *oides* means similar, shape, form or resembling (used in taxonomy), therefore *Myroides* resembling perfume, this due to the sweet odor, like fruits, characteristics in this genus [5]. It is an aerobic strict, thus is positive for oxidase and catalase activity, Gram-negative rod, 0.5 x 1–2 μm, non-motile, including gliding. Growing conditions characterizes its ability to grow on MacConkey agar and non-hemolysis production, growth at 18–22˚ C, and 37˚ C. *Myroides* produces, in the most of times, yellow or orange colonies [5].

Clinically, only four species of *Myroides* (*M. odoratimimus*, followed by *M. odoratus. M. injenensis* and *M. phaeus*) have been related with different infections such as endocarditis, pericarditis, urinary tract infections, skin and soft tissue infections, ventriculitis, liver abscesses and bacteremia, either in immunocompromised or immunocompetent host, as well [6–9]. The number of reported infections by *Myroides* species have been documented in a major frequency due the employment of more sensible and reliable techniques, such as matrix-assisted laser desorption/ionization time-of-flight mass spectrometry (MALDI-TOF MS) and 16S ARN sequencing; *Myroides* species have been reported specially in hospital outbreaks causing catheter related urinary tract infections [10, 11] associate, probable, to some factors such as prolonged use of antibiotics, immunosuppression, and invasive procedures [12]. Furthermore, it is linked to multiple antimicrobial resistance mechanisms [13]. However, as data expands, there must be more species associated with different infections that might be added, in this

sense, information, either microbiological or genetic, is limited and impact of *Myroides spp.*, infections are not well understood, therefore, the role of infections is poorly documented. The aim of this work was to describe the microbiological and genetic characteristics of seven different *Myroides spp.* clinical strains and comment on their phylum, pathogenic and resistance characteristics.

## Material and methods

### Clinical strains

We conducted a retrospective study seeking *Myroides spp.*, isolates from a tertiary care Hospital in Mexico City. We included all strains isolated from invasive samples, including blood cultures, lower respiratory samples, abscesses, and tissue biopsies, over an eleven-year period from 1/January/2012 to 1/January/2023 and were accessed for research on 15/May/2023. Duplicated samples were excluded. The collection and sample processing were conducted at Clinical Microbiology Laboratory at Instituto Nacional de Rehabilitación Luis Guillermo Ibarra Ibarra in Mexico City. All isolates were stored at -70˚C and subsequently inoculated onto 5% sheep blood agar prior to their use. Identification at the beginning was performed with the Vitek 2 compact (BioMérieux, Marcy ′Étoile, France) and was confirmed with Vitek MS (MALDI-ToF, (BioMérieux, Marcy ′Étoile, France)) following manufacturer's recommendations. Information from clinical records was obtained. Current work only involves clinical strains obtained by the typical procedures and guidelines, however, the strains used, and the experimental plan was approved by the research committee of the Institute with the assigned number 55/22 AC.

### Minimal inhibitory concentrations determination

The antibiotics evaluated were: Amikacin (AK), Gentamicin (GN), Aztreonam (ATM), Ceftazidime (CAZ), Cefepime (FEP), Ciprofloxacin (CIP), Levofloxacin (LVX), Meropenem (MEM), Imipenem (IMP), Colistin (CL), Piperacillin/Tazobactam (TZP), azithromycin (AZT), Erythromycin (E), Doxycycline (DO), Tigecycline (TYG) and Sulfamethoxazole/trimethoprim (SXT) (Sigma Aldrich, Burlington, Massachusetts, USA). Broth microdilution assay was performed following the recommendations by the CLSI [14]. *E. coli* ATCC 25922 was used for quality control. Due breakpoints are not defined for *Myroides spp.*, neither in CLSI or EUCAST MIC was defined as the lowest concentration that inhibits bacterial growth. For AK, GN, ATM, CAZ and FEP concentrations tested were from 256 µg/mL to 0.25 µg/m, while for CIP, LVX, MEM, IMP, AZT, E, DOX, and TYG concentrations were 64 µg/mL to 0.0612 µg/mL. On the other hand, for COL and TZP were 128 µg/mL to 0.125 µg/mL and 128/4 µg/mL to 0.125/4 µg/mL respectively. Finally, SXT starting concentration 8/152 µg/mL to 0.00078/0.1484 µg/mL.

### Phenotypic test to determine carbapenemase activity

Carbapenemase test was performed for each sample according to CLSI M100 2023 recommendation [14]. First, modified carbapenem inactivation method (mCIM) was carried oud, briefly, colonies were harvested with a 10 µL loop and deposited into a 2 mL polypropylene tube with 1.5 mL Trypticase soy broth, then a 10 µg MEM disc added into the tube. The tubes were incubated during 4 h at 37˚ C. After that, *E. coli* ATCC 25922 pan susceptible at 0.5 McFarland Scale was inoculated onto Mueller-Hinton plates. The same strain of *E. coli* ATCC 25922 was used as negative control and *Klebsiella pneumoniae* ATCC BAA-1705 as positive control (carbapenemase producer). The interpretation of the test was according to CLSI M100 2023 [14].

Due this method is validated just for *Enterobacterales* and *Pseudomonas aeruginosa*, a second method was performed, CARBA NP, also referred in CLSI M100 2023, briefly, two 600 μL conic tubes were labelled for each strain tested, then 100 μL of protein extraction buffer were added to each tube (a and b) and vortexed for 5 seconds and subsequently 100 μL of solution A (without IMP as substrate) and B (supplemented with IMP as substrate to carbapenemase) were added to the respective tubes labelled and again vortexed and incubated at 35° C/2 hours. Positive tests showed a change in color (yellow as positive and unchanged as negative). *E. coli* ATCC 25922 was used as negative control and *K. pneumoniae* ATCC BAA-1705 as positive control. The interpretation of the test was according to CLSI M100 2023 [14].

## Whole genome sequencing

Genomic DNA was extracted with DNeasy Blood & Tissue (QIAGEN). Once extracted, DNA quality and concentration were determined with the Qubit 3.0 fluorometer (Invitrogen, USA) and the Nanodrop One/One spectrophotometer (Thermo Fisher Scientific, USA) and samples were kept at -20°C, until use. Library preparation was performed using the Illumina DNA Prep (Illumina, USA) for the tagmentation, indexing and cleaning steps, as well as the employment of IDT for Illumina DNA/RNA UD Indexes (Illumina, USA) for the indexing step. Library quality control was performed using the Qubit 3.0 fluorometer (Invitrogen, USA) and the 4200 Tapestation System (Agilent, USA). Sequencing of pooled and normalized libraries was made using the MiSeq Reagent Kit V2 (300 cycles) on the Illumina MiSeq platform in a paired end configuration. Samples were sequenced at Centro Nacional de Referencia de Inocuidad y Bioseguridad Agroalimentaria from Servicio Nacional de Sanidad, Inocuidad y Calidad Agroalimentaria [15].

## Plasmid extraction and purification

In addition, plasmids were searched in all strains included. Plasmids were isolated and purified with the E.Z.N.A Plasmid DNA Mini Kit I (Omega bio-tek, Norcross, Georgia, USA). Once extracted, 5 μL of DNA were run in 1% agarose gel and visualized through Gel Doc XR+ with Image Lab Software (Bio-Rad; Hercules, California, USA), a 100 bp molecular ladder was used (Invitrogen; Waltham, Massachusetts, USA). Plasmid library preparation and sequencing were performed in the same manner as for genomic DNA [15].

## Genomic analysis

The Quality of the Sequenced Read Archives (SRA) generated was analyzed by using FastQC [16]. The removal of contaminations and sequences with poor Phred values <30 was conducted with Trimmomatic [17]. The SRA of each strain were assembled with SPAdes [18]. Each assembling quality were determined with QUAST [19]. The contigs of each genome were subject to the removal of sequences with coverage values <200 by using the seqtk_seq included in the sektq package [20]. A general annotation for the contigs was performed with Prokka [21]. The evaluation of the annotation based on completeness determination was conducted by using BUSCO [22].

A preliminary identification of the strains was performed with the generation of a phylogenetic tree based on the analysis of partial 16S rRNA. The sequences belonging to the 16S rRNA gene of each sequenced genome for this study were extracted by using barrnap [23]. The sequences of type strains were obtained from the non-redundant GenBank database [24]. A multiple sequence alignment (MSA) of 16S sequences was obtained with MUSCLE [25]. The Maximum Likelihood (ML) phylogeny was constructed in IQTREE employing the TVM + F + I + G4 evolutionary model with Ultrafast-Bootstrap determination of 10,000 replicates [26].

The 16S-based phylogenetic trees were edited and visualized with iTOL v5 [27]. As complementary, pairwise gene identity scores were determined with Protologger [28].

The annotation of Antimicrobial Resistance Elements (ARE) was conducted with Abricate [29], by using the Comprehensive Antibiotic Resistance Database (CARD) [30]. The annotation of the virulence factors (VF) was performed by using Markov Hidden Models with HMMER package [31] and custom scripts, based on the VF identified for *M. odoratimimus* [32]. The plasmids were extracted and typed with MOB Suite [33]; annotated and visualized with Proksee web server [34].

## Results

### Strains included

During the study period, a total of seven strains previously identified such as *Myroides spp.*, which were confirmed with Vitek 2 compact and with Vitek MS, such as *Myroides* spp. Four isolates affected males, five cases were associated with skin and deep tissue infections as showed in Table 1.

Treatment chosen was different for each strain including fluoroquinolones, aminoglycosides and carbapenems, meanwhile for osteomyelitis the antimicrobial selected was meropenem, with different time periods depending on the type of infection.

### Susceptibility patterns

The greatest resistance rates were observed for aminoglycosides (AK and GN) with MICs $\geq$ 256 µg/mL, followed by ß-lactams (ATM, cephalosporins and TZP). On the other hand, azithromycin had lower MICs (around 2 µg/mL the most) in comparison with erythromycin which had the higher concentration with 4 µg/mL and the lowest with 0.5 µ/mL; however, tetracyclines showed similar MICs between members assessed. The lowest MICs observed in antibiotics evaluated were in SXT as reported in Tables 2 and 3.

### Phenotypic test to determine carbapenemase activity

Due to high MICs values for ß-lactam antibiotics, including carbapenem, we performed phenotypic assay looking for carbapenemases presence according to CLSI guidelines M100, however, this procedure is not yet validated and standardized for microorganisms different to

**Table 1. Clinical characteristics, treatment given, and outcomes of clinical strains included.**

| Number | Gender/ Age (years) | Comorbidities | Type of infection | Clinical sample | Isolation Date | Treatment | Outcome |
|---|---|---|---|---|---|---|---|
| EB1487 | Male / 17 | Congenital scoliosis | Catheter related Urinary tract infection | Urine | 23/04/2014 | Ciprofloxacin | Cured |
| C1519 | Female / 37 | Spinal cord injury and trauma | Osteomyelitis | Bone biopsy | 06/08/2015 | Meropenem | Cured |
| C1996 | Female / 41 | Burn injury | Skin and Soft tissue infection | Quantitative biopsy (skin) | 14/11/2016 | Meropenem | Died / Other infections |
| C2723 | Male / 11 | Trauma | Skin and Soft tissue infection | Quantitative biopsy (skin) | 29/11/2018 | Amikacin | Cured |
| C4067 | Male / 8 | Trauma | Skin and Soft tissue infection | Quantitative biopsy (skin) | 15/07/2021 | Ciprofloxacin | Cured |
| C4256 | Male / 24 | Burn injury | Osteomyelitis | Bone biopsy | 28/09/2021 | Meropenem | Cured |
| C4411 | Female / 48 | Ependymoma | Catheter related urinary tract infection | Urine | 09/12/2021 | None (catheter changed) | Cured |

**Table 2. *Myroides spp*. minimum inhibitory concentrations of clinical strains.**

| Strain | AK (µg/mL) | GN (µg/mL) | ATM (µg/mL) | CAZ (µg/mL) | FEP (µg/mL) | CIP (µg/mL) | LVX (µg/mL) | MEM (µg/mL) | IMP (µg/mL) | COL (µg/mL) | TZP (µg/mL) |
|---|---|---|---|---|---|---|---|---|---|---|---|
| EB1487 | 256 | > 256 | 128 | 256 | 64 | 4 | 4 | 4 | 4 | > 128 | 128/4 |
| C1519 | > 256 | > 256 | 64 | 128 | 32 | 0.5 | 0.5 | 8 | 4 | > 128 | 128/4 |
| C1996 | > 256 | > 256 | 256 | 256 | 64 | 64 | 32 | 2 | 2 | > 128 | 128/4 |
| C2723 | 256 | 64 | 256 | 64 | 8 | 2 | 4 | 1 | 0.25 | > 128 | 32/4 |
| C4067 | > 256 | > 256 | 256 | 256 | 64 | 2 | 2 | 8 | 4 | > 128 | 128/4 |
| C4256 | > 256 | > 256 | 256 | 256 | 64 | 2 | 2 | 8 | 4 | > 128 | 128/4 |
| C4411 | > 256 | > 256 | 256 | 256 | 64 | 2 | 2 | 8 | 4 | > 128 | 128/4 |

AK: amikacin; GN: gentamicin; ATM: aztreonam; CAZ: ceftazidime; FEP: cefepime; CIP: ciprofloxacin; LVX: levofloxacin; MEM: meropenem; IMP: imipenem; COL: colistin; TZP: piperacillin/tazobactam

*Enterobacterales* or *Pseudomonas aeruginosa*. We identified hydrolytic activity against MEM in EB1487, C1519, C4411 with halos < 15 mm, on the other hand, C1996 and C4256 had halos with 16 mm, according CLSI M100 definitions, for *Enterobacterales* and *Pseudomonas aeruginosa* this measure corresponds with the definition of inconclusive. However, with CARBA NP strain C1996 was interpreted as positive as well as C4256. Previously C4067 was defined to be negative with mCIM however, once repeated with CARBA NP was defined as positive (S1 Fig). While the rest of the strains had halos > 19 mm (negative); however, as mentioned above, this is not a validated strategy for this microorganism, so the negative result could be due to a limitation of the strategy.

## Molecular identification

The phylogenetic tree reconstructed exhibited the following: four strains (C4366, EB1487, C4411 and C1996) were clustered with *M. odoratimimus* type strain, with an identity percentage > 99% (Fig 1), in Table 4 are included all those non-repeated sequences downloaded to build the phylogeny. Hence, the results indicate that these four strains may be identified as *M. odoratimimus*. By the other hand, the strain C4067 evidenced a phylogenetic relationship with *Myroides marinus*, however, they shared a 95.67% of identity. Likewise, the strains C2723 and C1519 were clustered in the same clade with *Myroides phaeus*, showing a 96.3% of identity among them; consequently, we suspect the strains of *Myroides* C4067, C2723 and C1519 may be cataloged as new species.

**Table 3. *Myroides spp*. minimum inhibitory concentrations.**

| Strain | AZT (µg/mL) | E (µg/mL) | DOX (µg/mL) | TIG (µg/mL) | SXT (µg/mL) |
|---|---|---|---|---|---|
| EB1487 | 1 | 4 | 0.5 | 2 | 1/19 |
| C1519 | 1 | 0.5 | 0.125 | 1 | 0.5/9.5 |
| C1996 | 2 | 4 | 2 | 2 | 1/19 |
| C2723 | 0.5 | 2 | 0.5 | 0.25 | 2/38 |
| C4067 | 2 | 2 | 1 | 0.5 | 1/19 |
| C4256 | 2 | 4 | 1 | 0.5 | 1/19 |
| C4411 | 2 | 2 | 1 | 0.5 | 2/38 |

AZT: azithromycin; DOX: doxycycline; E: erythromycin; TIG: tigecycline; SXT: sulfamethoxazole/trimethoprim.

Tree scale: 0.01 ⊢——⊣

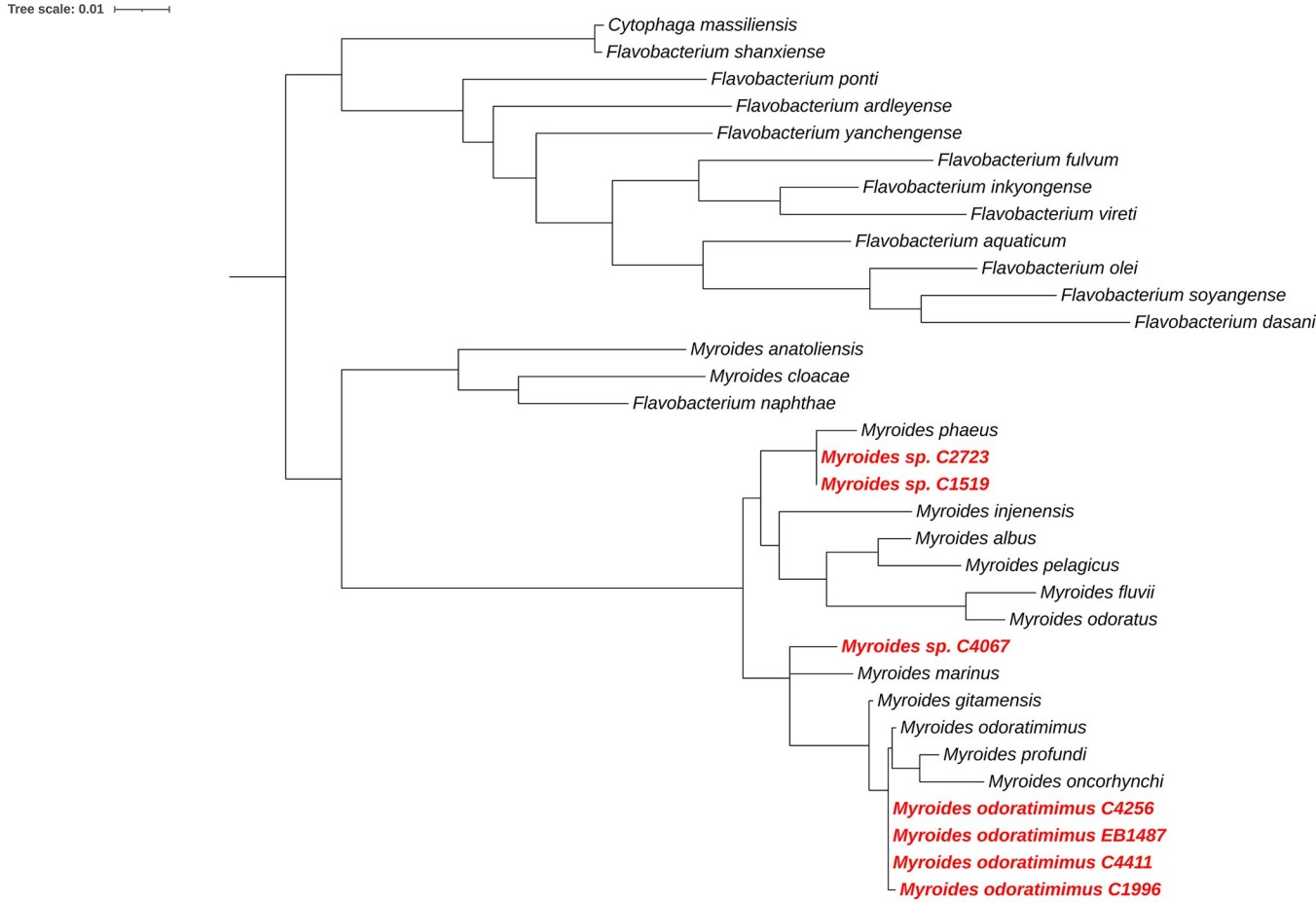

**Fig 1. ML phylogenetic tree of the 16S rRNA sequences.** The strains evaluated in this study are marked in bold blue color into the tree. Numbers on the nodes and branches represent the Ultrafast Bootstrap values of 10,000 replicates. Scale bar represents the number of nucleotide differences between branches.

## Virulence Factors (VF) and Antimicrobial Resistance (AMR) genes detection

A total of 15 elements associated with VF and classified in seven categories were recognized and annotated. The summary of this result is displayed in Fig 2. All elements were identified in all strains, nevertheless *sodB* (encoding for Superoxide dismutase) was not identified in the isolates C1519 and C2723, whereas *wecC*, encoding for UDP-N-acetyl-D-mannosaminuronic acid dehydrogenase was not detected in the isolate EB1487. The VF recognized maybe associated with production of capsular polysaccharide, cell wall peptidoglycan, heme biosynthesis (oxidative respiration), intracellular survival factors (ISF), chaperones, streptococcal enolase and type 4 secretion system effectors (T4SS effectors).

Also 8 kinds of elements associated with Antibiotic Resistance (ARE) were detected (Fig 3). The strains showed an average of about 15 AMR genes, exhibiting an evident multidrug resistance and harboring high diversity of AREs. The isolate *Myroides spp*. C4067 presented the highest diversity and content of AREs, with a total of 22 genes. On the other hand, the strain *Myroides spp*. C2723 exhibited the lowest content of AMR genes. In general, the drug class assigned for the AMR genes identified was congruent with the MIC profiles obtained in experimental assays. Even though the *tet(X2)* gene was identified in two strains, these displayed

**Table 4. Sequences of study for ML phylogenetic tree reconstruction.**

| Organism | GenBank ID |
| --- | --- |
| *Myroides odoratus* | M58777 |
| *Myroides fluvii* | MK129421 |
| *Myroides pelagicus* | AB176662 |
| *Myroides albus* | MK734183 |
| *Myroides injenensis* | HQ671078 |
| *Myroides phaeus* | GU253339 |
| *Myroides oncorhynchi* | OL437262 |
| *Myroides profundi* | EU204978 |
| *Myroides odoratimimus* | AJ854059 |
| *Myroides gitamensis* | HF571338 |
| *Myroides marinus* | GQ857652 |
| *Flavobacterium olei* | KX672808 |
| *Flavobacterium azizsancarii* | OQ024221 |
| *Flavobacterium soyangense* | KX061439 |
| *Flavobacterium dasani* | MH019224 |
| *Flavobacterium aquaticum* | HE995762 |
| *Flavobacterium vireti* | KM576853 |
| *Flavobacterium inkyongense* | KX025140 |
| *Flavobacterium fulvum* | KU052686 |
| *Flavobacterium yanchengense* | JX548325 |
| *Flavobacterium ardleyense* | KX911209 |
| *Flavobacterium aquimarinum* | KY612936 |
| *Flavobacterium ponti* | GQ370387 |
| *Flavobacterium shanxiense* | KJ641612 |
| *Cytophaga massiliensis* | EF394924 |
| *Flavobacterium naphthae* | MF405102 |
| *Myroides cloacae* | KU746272 |
| *Myroides anatoliensis* | JF825522 |

susceptibility to tetracyclines. Also, the *iri* and *rphA*, linked with rifampicin resistance were detected. The strains exhibited a significant abundance of AREs associated with ß-lactam, aminoglycosides, and glycopeptides resistance, matching with the resistance profiles to this antibiotic observed. Interestingly, all strains showed the presence of *ereB*, an erythromycin esterase-like associated with resistance to macrolides, despite the strains being susceptible to azithromycin. Interestingly, strains with GOB-16, a metallo ß-lactamase, were C4256, EB1487, C4067, C4411, these same strains shared another metallo ß-lactamase belonging to the IMP family (IMP-27), strain C1996 carried IMP-27 but not GOB-16. On the other hand, strains C4411 and C4256 co-carried two IMPs (IMP-27 and IMP-35) in addition to GOB-16. MUS-1 is a metallo ß-lactamase identified in *Myroides* present in almost all isolates except C2723 and C1519. Of the members of the oxacillinase (OXA) family with carbapenemase activity we found OXA-229 (C4411, C4067 and C4256), OXA-351 (C4067) and OXA-97 (C4256). Of the AmpC group, PDC-85 was found in C2723 and C1519. Genes associated with colistin resistance were found in *mcr3.10*, *mcr3.6* and *mcr3.7*. Strains C1996, C4256, C4067 shared the presence of *mcr3.6* and *mcr3.7* genes. EB1487 co-carried *mcr3.10* and *mcr3.6* while C4411 carried only *mcr3.7*. Phenotypically, EB1487, C1519 and C4411 showed carbapenemase activity against meropenem.

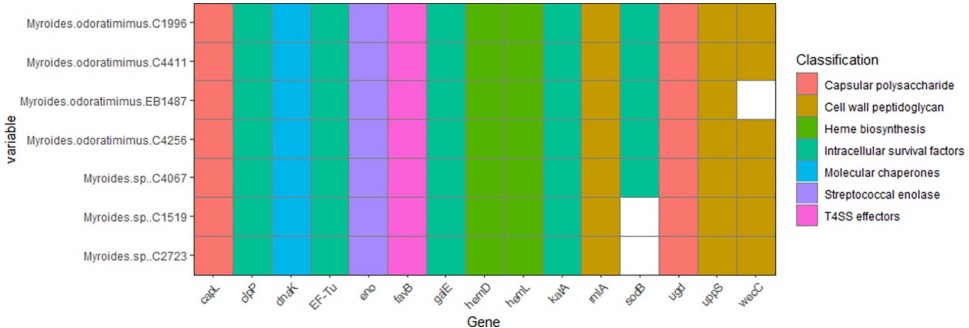

**Fig 2. Virulence factors found in clinical strains of *Myroides* spp.**

## Plasmid identification and annotation

Two plasmids were detected (S2 Fig) in the strains *Myroides* spp. C2723 (Plasmid ID C2723_C6) (Fig 4) and *M. orodatimimus* EB1487 (Plasmid ID EB1487_S7) (Fig 5). The C2723_C6 showed an approximate size of 19 Kb, whereas EB1487_S7 exhibited a size of 150 Kb. Therefore, we consider EB1487_S7 as a megaplasmid. The annotation of the plasmid showed that C2723_C6 harbors ribosomal RNA genes, as well as tRNA sequences. A total of 11 proteins were predicted, however, the annotation assigned them as hypothetical proteins.

By the other hand, the EB1487_S7 also harbored ribosomal RNA and tRNA genes, and 101 proteins were predicted, of which 75 were cataloged as hypothetical proteins. Twenty-six genes were properly annotated for EB1487_S7 whose functional classification may be associated with mobile elements, DNA processing, amino acid metabolism, aerobic respiration, cell wall peptidoglycan biosynthesis, carbon metabolism, ROS defense, and membrane transport.

## Discussion

Since the reclassification of *Flavobacterium odoratum* and the creation of the new genus *Myroides* by Dr. Vancanneyt M et al. in 1996 [4], very little evidence has been generated on the role of *Myroides* associated with infections in patients, only 43 articles were found in PubMed since 2000 using the search terms "*Myroides*" and "infection". Of these, bacteremia and urinary tract infection were the most frequent. The most common species were *M. odoratimimus* and *M. odoratus* [6–8, 10, 11, 35–71]

*Myroides* species are typically found in immunocompromised hosts, although there have been a few cases reported in immune competent individuals [6, 68, 69]. In our series, skin, and soft tissue infections, as well as deep infections, were the most common. This is likely since we

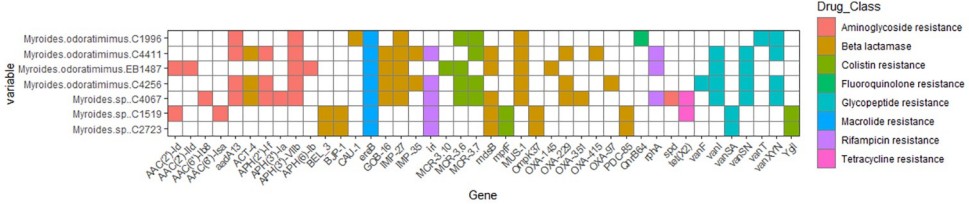

**Fig 3. Antimicrobial resistance elements presents in *Myroides* species strains.**

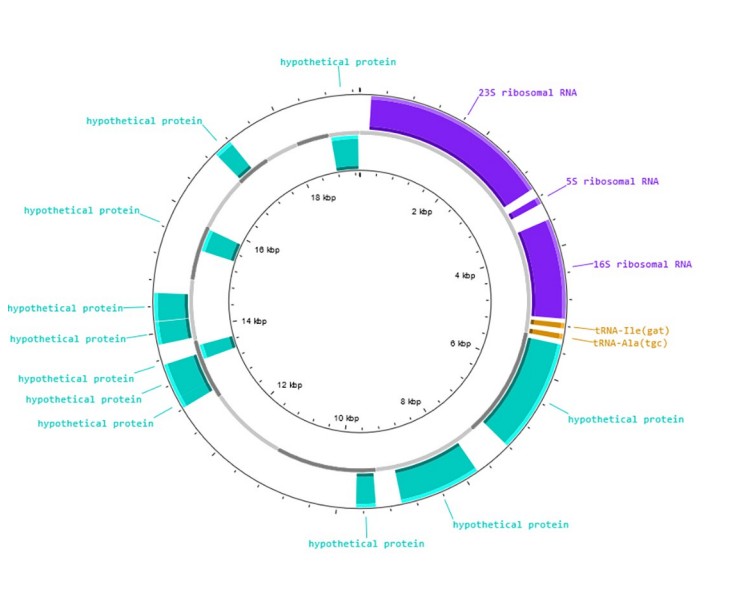

**Fig 4. Genomic map of the identified plasmid pM2723.** The inner circle represents the genome size per region. The outer circle marks all open reading frames and proteins predicted. Forms and letters in purple show the location of ribosomal RNA genes. Forms and letters in orange indicate the location of tRNA genes. Forms and letters in blue indicate the location of annotated and unannotated proteins.

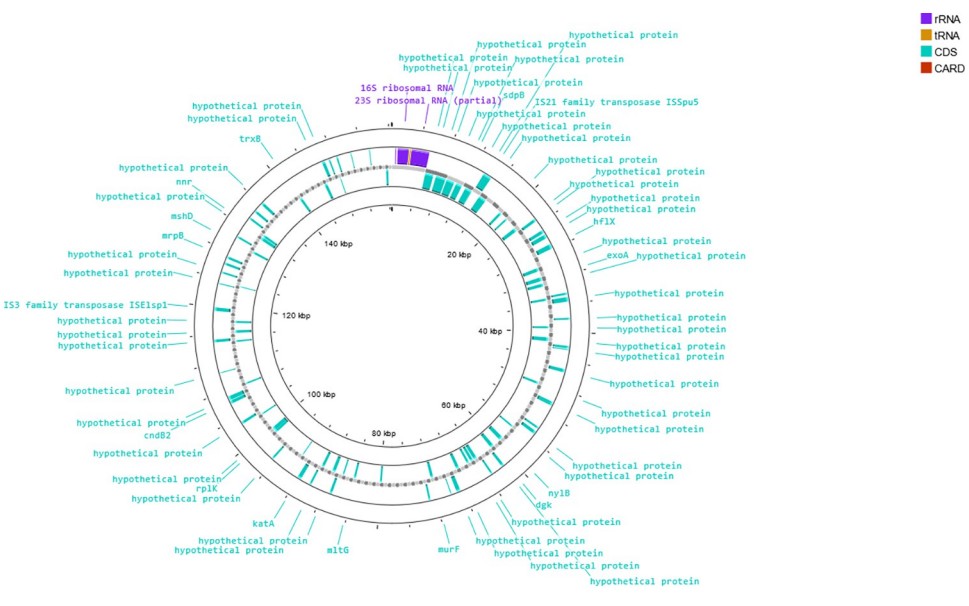

**Fig 5. Genomic map of the identified plasmid EB1487_S7.** The inner circle represents the genome size per region. The outer circle marks all open reading frames and proteins predicted. Forms and letters in purple show the location of ribosomal RNA genes. Forms and letters in orange indicate the location of tRNA genes. Forms and letters in blue indicate the location of annotated and unannotated proteins.

are a tertiary care hospital specializing in orthopedics, rehabilitation, and burns. Except for one patient, all were successfully treated despite having various comorbidities. The deceased patient had burn injuries and was infected with *Myroides*, along with other Gram-negative bacteria. Burn patients are severely immunosuppressed, making them more susceptible to infections due to the loss of their skin barrier. No previous cases of *Myroides* infection have been reported in burned patients.

The identification of rutinary cultures within a clinical microbiology lab is performed with miniaturized biochemistry systems such as Vitek 2 [72] (BioMérieux, Marcy 'Étoile, France) and Phoenix (Becton-Dickinson, New Jersey, USA) [43]. The introduction of instruments with larger databases and better precision and sensitivity, such as MALDI-ToF, has increased detection capacity, but even these types of systems, have methodological limitations for a suitable discrimination among species [73, 74]. Therefore, we recurred to a presumptive molecular identification of pathogens based on phylogenetic analysis of 16S rRNA. The phylogenetic reconstruction has been useful for other studies for resolving properly the identity of isolates belonging to *Stenotrophomonas* genus [15]. The use of 16S analysis as a robust tool for identification in the clinical routine is not common [75], but has been widely explored in clinical research studies for a presumptive assignment of identity to bacteria species [76–79]. According to the criteria for assignment of bacterial species established by Roselló Mora [80], based on 16S-rRNA identity percentage (>97%). We proposed that four isolated species were designated as *M. odoratimimus*, while the strains C2723, C1519 and C4067 may be cataloged as new species of *Myroides*. The phylogenetic evidence as a robust tool for identification supports our predictions. Nonetheless, further studies based on a more rigorous genome-based phylogenetic analysis are needed to validate our hypothesis.

Due to the broad AMR spectrum observed in the isolates of study, we decided to identify the ARE presumptively involved in pan-drug resistance. For instance, previous reports describe that *M. odoratimimus* may harbor up to 32 AMR genes [32, 81, 82] of which ß-lactamases and efflux pumps are the most abundant AREs observed. From 2014 until now, the main AREs identified for *Myroides* spp., has been the following: ß-lactamases: $bla_{VIM}$, $bla_{IMP}$, $bla_{NMD}$, $bla_{OXA-78}$, $bla_{OXA-209}$, $bla_{OXA-347}$, $bla_{KPC}$, $bla_{TUS}$, $bla_{EBR-1}$, $bla_{MOC}$ and $bla_{MUS-1}$; tetracycline resistance: *tet(X6)*; efflux pumps: *abeS, msrB, qacH, rosA*; macrolides: *erm(F), ere(D)* [12, 32, 55, 83–85]. Here, we were able to detect the AREs aforementioned; however, it is important to highlight that the isolates analyzed in this work also displayed AREs and a phenotype associated with resistance to colistin, glycopeptides and rifampicin, a resistance trait not observed previously. These results suggest that the resistome of *Myroides* spp. is expanding remarkably, and probably soon the species of this genus might convert into an important health threat.

Other important trait is that multidrug resistance genotype was observed in the presumptive new species, although only *M. orodatimimus*, *M. odoratus*, *M. phaeus* and *M. injerensis* have been cataloged as relevant AMR pathogens [45, 57, 82, 83], our results indicate that the multidrug resistance phenotype possibly is highly expanded and extended in several species of *Myroides*. In the case of *M. orodatimimus* a chromosomic metallo ß-lactamase has been describe, the MUS-1 [86]. Two strains had an undetermined phenotype, however carrier, this due the phenotypic assay is not validated for bacteria different to *Enterobacterales* or *Pseudomonas* spp,

Pathogen *Myroides* species have been shown to possess a number of VF that are associated with a range of processes, including cell adherence, intracellular survival, and capsule production [32, 82]. Nevertheless, a more comprehensive investigation into the molecular mechanisms underlying the pathogenicity of *Myroides* spp. has yet to be conducted. Therefore, the current state of knowledge regarding the infection process and disease-causing mechanisms of *Myroides* is limited. Based on the identified VF, the results suggest that the isolates under study

possess the ability to survive and replicate within host cells, evade phagocytic cells of the immune system, and are able to cause disease. This is a virulence trait observed in other opportunistic pathogens, including *Mycobacterium* spp., *Pseudomonas* spp., *Serratia* spp., and others [87–89].

Certainly, our work has limitations, design nature without doubts since is centered only the background of clinical strains of *Myroides* spp., the number of strains included, in this sense information about prevalence around the world is limited beside we included all references published. The correlation with a *in vivo* model could evaluate the impact of virulence factors and the need of phenotypic methods to explore the activity of carbapenemases in this genus.

## Conclusions

The *Myroides* genus is a rare microorganism that can cause infections in both immunosuppressed and immune competent hosts. The outcome of infection varies depending on the antimicrobial profile of the bacteria. Molecular techniques are essential for accurately identifying these bacteria, characterizing potential species, and exploring therapeutic options for achieving clearance of infections. More studies are needed to prove that strains C2723, C1519 and C4067 are new species within the *Myroides* genus. Finally, the species analyzed in the present work showed reduced susceptibility patterns for aminoglycosides, ß-lactams, cyclic lipopeptides, this was corroborated with the genes found in the genome, therefore, susceptibility tests should be performed in the clinical microbiology laboratory and the Infectious diseases specialists staff should consider these susceptibility patterns for the treatment scheme of patients suffering infections by microorganisms of this genus.

## Supporting information

**S1 Fig. CARBA NP in *Myroides spp*. clinical strains.**
(DOCX)

**S2 Fig. Plasmids extracted from *Myroides spp*. clinical strains.** MWM: Molecular weight marker.
(DOCX)

**S1 Raw image. Plasmids extracted from *Myroides spp*. clinical strains.** MWM: Molecular weight marker. (Raw data, letters were added with the transilluminator software).
(DOCX)

## Acknowledgments

To SENASICA and WHO Collaborating Centre on Antimicrobial Resistance in Foodborne and Environmental Bacteria (MEX-33), especially Mayrén Zamora Nava and all the sequencing and bioinformatics staff for helping us to sequence our samples.

## Author Contributions

**Conceptualization:** Jossue Mizael Ortiz-Álvarez, Rafael Franco-Cendejas, Luis Esaú López-Jácome.

**Formal analysis:** Claudia Adriana Colín-Castro, Jossue Mizael Ortiz-Álvarez, Cindy Fabiola Hernández-Pérez, Melissa Hernández-Durán, María de Lourdes García-Hernández, María Guadalupe Martínez-Zavaleta, Noé Becerra-Lobato, Mercedes Isabel Cervantes-Hernández, Graciela Rosas-Alquicira, Guillermo Cerón-González, Braulio Josué Méndez-Sotelo, Rodolfo García-Contreras, Rafael Franco-Cendejas, Luis Esaú López-Jácome.

**Investigation:** Claudia Adriana Colín-Castro, Jossue Mizael Ortiz-Álvarez, Cindy Fabiola Hernández-Pérez, Melissa Hernández-Durán, María de Lourdes García-Hernández, María Guadalupe Martínez-Zavaleta, Noé Becerra-Lobato, Mercedes Isabel Cervantes-Hernández, Graciela Rosas-Alquicira, Guillermo Cerón-González, Braulio Josué Méndez-Sotelo, Rodolfo García-Contreras, Luis Esaú López-Jácome.

**Methodology:** Claudia Adriana Colín-Castro, Jossue Mizael Ortiz-Álvarez, Cindy Fabiola Hernández-Pérez, Melissa Hernández-Durán, María de Lourdes García-Hernández, María Guadalupe Martínez-Zavaleta, Noé Becerra-Lobato, Mercedes Isabel Cervantes-Hernández, Graciela Rosas-Alquicira, Guillermo Cerón-González, Braulio Josué Méndez-Sotelo, Rodolfo García-Contreras, Luis Esaú López-Jácome.

**Supervision:** Rafael Franco-Cendejas, Luis Esaú López-Jácome.

**Writing – original draft:** Claudia Adriana Colín-Castro, Jossue Mizael Ortiz-Álvarez, Rodolfo García-Contreras, Rafael Franco-Cendejas, Luis Esaú López-Jácome.

**Writing – review & editing:** Claudia Adriana Colín-Castro, Jossue Mizael Ortiz-Álvarez, Rodolfo García-Contreras, Rafael Franco-Cendejas, Luis Esaú López-Jácome.

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
