## [Decision Letter · Decision Letter 0]

17 Jul 2024

PONE-D-24-22192Myroides species, pathogenic spectrum and clinical microbiology sight in Mexican isolates.PLOS ONE

Dear Dr. Lopez Jacome,

Thank you for submitting your manuscript to PLOS ONE. After careful consideration, we feel that it has merit but does not fully meet PLOS ONE’s publication criteria as it currently stands. Therefore, we invite you to submit a revised version of the manuscript that addresses the points raised during the review process.

We look forward to receiving your revised manuscript.

Kind regards,

Mabel Kamweli Aworh, DVM, MPH, PhD. FCVSN

Academic Editor

PLOS ONE

3. We notice that your supplementary figures are uploaded with the file type 'Figure'. Please amend the file type to 'Supporting Information'. Please ensure that each Supporting Information file has a legend listed in the manuscript after the references list.

Additional Editor Comments:

In addition to addressing all the comments of the reviewers' kindly address the following;

1. Lines 120-124: Please provide the range of concentrations for each antimicrobial tested.

2. Lines 362–368: This is not clear. Please rewrite for clarity and flow.

3. Highlight the key limitations of the present study in the last paragraph of the discussion section.

4. In your conclusion, can you provide some recommendations based on your study findings?

Reviewers' comments:

Reviewer's Responses to Questions

**Comments to the Author**

1. Is the manuscript technically sound, and do the data support the conclusions?

Reviewer #1: Yes

Reviewer #2: Yes

Reviewer #3: Partly

2. Has the statistical analysis been performed appropriately and rigorously? 

Reviewer #1: N/A

Reviewer #2: Yes

Reviewer #3: Yes

3. Have the authors made all data underlying the findings in their manuscript fully available?

Reviewer #1: Yes

Reviewer #2: Yes

Reviewer #3: Yes

4. Is the manuscript presented in an intelligible fashion and written in standard English?

Reviewer #1: Yes

Reviewer #2: Yes

Reviewer #3: Yes

5. Review Comments to the Author

Reviewer #1: This is a good manuscript with impressive efforts from the authors. A couple of areas require further clarification and possible additions

line 48 - The introduction in the abstract does not clearly state the significance of this study. What is the public health impact of understanding the pathogenic spectrum or clinical microbiology of Myroides species in Mexico? This needs to be clearly stated here.

line 55 - Indicate the study design that was used here. Was this a retrospective analysis? Also, indicate the period of the specimen was collection. This makes the methods clearer

line 57 - include total number of myroides isolates recovered or used and the percentage of species of particular type

line 58 - This statement makes it look like you identified only 6 instead of 7 strains highlighted in the main paper. can you clarify?

line 61 - Are we implying that 100% of isolates were resistant to all these antibiotics mentioned? Its not very clear from this sentence. If other otherwise, please state what percentage of isolates were resistant to antibiotic A and what percentage was resistant to antibiotic B e.t.c

line 96 - It will be interesting to know, any specific concerns or knowledge gaps regarding Myroides species in Mexico? Are there any existing studies on their prevalence or clinical impact in this region? it is not clear if you have done an effective review of relevant existing literature on Myroides species and their clinical significance, particularly in a Mexican context.

lines 102 - 190 - No ethical considerations were mentioned in the methods. Were there any ethical considerations related to the use of clinical specimens?

lines 102 - 190 - what data collection/recording tool was used to collect or record the data on characteristics of patients e.g Gender, Age, Comorbidities, Type of infection, Clinical sample Isolation, Date Treatment Outcome e.t.c? Did you use a proforma? was this a standard tool? I cant seem to find any details in the methods for this. please indicate

line 205 - Table 1 - Considering that this study has to do with Myroides species which are typically found in immunocompromised hosts, did you consider collecting or recording data on the immune status of the patients? I know you had a lot of burns patients. But could the this have anything to do with the treatment and outcome of these patients outlined in table 1?

lines 306 - 368 - what are the limitations of this study? I am not able to clearly see where you highlighted the limitations.

Since the phenotypic assay you used is not validated for the specific microorganisms tested (outside of Enterobacterales and Pseudomonas aeruginosa), can you elaborate on the potential limitations of using this assay for these particular strains?

- For strains C1996 and C4256 with 16 mm halos, which falls under the inconclusive range according to CLSI M100 for Enterobacterales and Pseudomonas aeruginosa, how do you plan to address this inconclusiveness? Are there any additional confirmatory tests planned to definitively determine the presence or absence of carbapenemase activity in these strains?

- Given the limitations of the current assay, would exploring alternative methods for carbapenemase detection, such as molecular techniques (e.g., PCR), be a valuable addition to strengthen the study's findings?

- Can you suggest any modifications or optimizations to the current phenotypic assay that might improve its applicability to a broader range of microorganisms?

How do you plan to address the limitations of the current approach in future studies investigating carbapenemase activity in these specific microorganisms?

Reviewer #2: The authors described intelligently the materials and methods used in the research. The research results were reported with clarity devoid of ambiquity.

Areas of further research need to be succintly stated under the conclusion section in the manuscript. There are mentions of additional research in the manuscript in line 337 - 338 and 362-363; authors are advised to state them in the conclusion section.

Authors should provide study limitations in the manuscript

A typo/unclear statement in line 371. Authors are advised to edit this typographical error.

Reviewer #3: Myroides species, pathogenic spectrum and clinical microbiology sight in Mexican isolates.

Reviewer’s comments

• The Introduction was well-written with clear objectives, the gap identified and appropriate links given.

• The Methods used were appropriate for the set objectives.

• The results were accurately presented and the discussion derived from the findings and reasonable inference drawn.

• The analysis of the results were scientific and exact.

Accept after minor revisions.

Lines 137-149

Author should cite a reference for the method used for whole genome sequencing.

Lines 150-156

• Authors should cite a reference for the method used here too.

• Methods should be well cited.

Lines 371-371

• Myroides genus is not such as uncommon microorganism causing infections, both in

• immunosuppressed and immune competent hosts, with different outcomes depending on

• antimicrobial profiles. WHY NOT SAY: Myroides genus is a common microorganism causing...

Page 7-40

• Numbering is not appropriate. Need to check the numbering and number the required lines appropriately.

Page 15

• Are there more specific ways to itemize the data availability Statement.

• Ensure regular spacing through the write up.

In conclusion, authors should identify any limitations to the execution of the research as this is necessary and obvious going by the methods used.

6. PLOS authors have the option to publish the peer review history of their article (what does this mean?). If published, this will include your full peer review and any attached files.

Reviewer #1: **Yes: **Abdulhakeem Abayomi Olorukooba

Reviewer #2: No

Reviewer #3: **Yes: **Taiwo Akindahunsi

---

## [Author Response · Author response to Decision Letter 0]

12 Aug 2024

Rebuttal letter

First of all, we would like to thank the reviewers for their time and comments on this paper.

Reviewer 1. 

Line 48 - The introduction in the abstract does not clearly state the significance of this study. What is the public health impact of understanding the pathogenic spectrum or clinical microbiology of Myroides species in Mexico? This needs to be clearly stated here.

Thanks a lot for the comment, we have added the next to your consideration. “In addition, their pathogenic role is limited in terms of reporting their microbial physiology, so the present work provides information in this regard in addition to the information that is available in the international literature.”

In this sense, information in our work will not be exclusive to the Mexican microbiological community, but will give evidence to the international scientific community on aspects of susceptibility and virulence factors.

Line 55 - Indicate the study design that was used here. Was this a retrospective analysis? Also, indicate the period of the specimen was collection. This makes the methods clearer.

Line 57 - include total number of myroides isolates recovered or used and the percentage of species of particular type

Complete description about the time period was mentioned at material and methods, however, thanks to your observation we have added “Methods: Seven Myroides spp., strains associated with infections were included from 1/January/2012 to 1/January/20 and identified by miniaturized biochemistry and MALDI-ToF”. 

Line 58 - This statement makes it look like you identified only 6 instead of 7 strains highlighted in the main paper. can you clarify?

I so sorry about that, We have added the number of strains, were in total 7, as is mentioned in objective, and methods. Sorry for the confusion.

Line 61 - Are we implying that 100% of isolates were resistant to all these antibiotics mentioned? Its not very clear from this sentence. If other otherwise, please state what percentage of isolates were resistant to antibiotic A and what percentage was resistant to antibiotic B e.t.c.

Thank you very much for your kind and proper observation. Resistance leve lis mentioned in results due the characters numbers for abstract. However, we added the next: against these antibiotics (AK and GN both 100%, ATM, CAZ and FEP 100%, e.g.); moreover, the presences of carbapenemases were evidenced by meropenem (mCIM) and imipenem (CARBA NP) degrading activity in six isolates and” Thank you.

Line 96 - It will be interesting to know, any specific concerns or knowledge gaps regarding Myroides species in Mexico? Are there any existing studies on their prevalence or clinical impact in this region? it is not clear if you have done an effective review of relevant existing literature on Myroides species and their clinical significance, particularly in a Mexican context

This is a very important comment and thank you very much for your question. There is no information about prevalence in our country o region, as we mentioned in Discussion, we performed research on pubmed since 2020 and we just found 43 papers related with Myroides causing infection and all those papers were properly cited in our work, besides the most of those Works are case reports, in our knowledge (with humility) this is the first in our country.

Lines 102 - 190 - No ethical considerations were mentioned in the methods. Were there any ethical considerations related to the use of clinical specimens?

Thank you for this observation, we only used strains, and was conducted according to research acceptance with assigned ID 55/22 AC from our institution. 

Lines 102 - 190 - what data collection/recording tool was used to collect or record the data on characteristics of patients e.g Gender, Age, Comorbidities, Type of infection, Clinical sample Isolation, Date Treatment Outcome e.t.c? Did you use a proforma? was this a standard tool? I cant seem to find any details in the methods for this. please indicate

Data requested is from electronical record and it is mentioned in table 1. Proforma was not used. 

Line 205 - Table 1 - Considering that this study has to do with Myroides species which are typically found in immunocompromised hosts, did you consider collecting or recording data on the immune status of the patients? I know you had a lot of burns patients. But could the this have anything to do with the treatment and outcome of these patients outlined in table 1?

Actually this a great idea and research question for future studies, but due to the nature of design we want to know about the genetic background of Myroides, what if more microbiologist isolate this bacteria and to decide give certain antibiotics, if we give information about resistance genes, susceptibility behavior could support microbiologist and infectious diseases specialist, probably in other kind of patients or institutions this bacteria are mos frequent that with us, we don’t know about that. Then immunocompromised status and those factors that may influence host vulnerability was not explored. Not in this time.

lines 306 - 368 - what are the limitations of this study? I am not able to clearly see where you highlighted the limitations.

About this point we have added the next: Certainly, our work has limitations, design nature without doubts since is centered only the background of clinical strains of Myroides spp., the number of strains included, in this sense information about prevalence around the world is limited beside we included all references published. The correlation with a in vivo model could evaluate the impact of virulence factors and the need of phenotypic methods to explore the activity of carbapenemases in this genus.

Since the phenotypic assay you used is not validated for the specific microorganisms tested (outside of Enterobacterales and Pseudomonas aeruginosa), can you elaborate on the potential limitations of using this assay for these particular strains?

Was added according to your proper recommendation. the need of phenotypic methods to explore the activity of carbapenemases in this genus.

- For strains C1996 and C4256 with 16 mm halos, which falls under the inconclusive range according to CLSI M100 for Enterobacterales and Pseudomonas aeruginosa, how do you plan to address this inconclusiveness? Are there any additional confirmatory tests planned to definitively determine the presence or absence of carbapenemase activity in these strains?

Thank you very much, and again thanks to your comments we decided to include another experiment in order to prove the carbapenemase activity, in this case we change the substrate (IMP instead up MEM) and the used was CARBA NP. We added the next. Line 136 Due this method is validated just for Enterobacterales and Pseudomonas aeruginosa, a second method was performed, CARBA NP, also referred in CLSI M100 2023, briefly, two 600 µL conic tubes were labelled for each strain tested, then 100 µL of protein extraction buffer were added to each tube (a and b) and vortexed for 5 seconds and subsequently 100 µL of solution A (without IMP as substrate) and B (supplemented with IMP as substrate to carbapenemase) were added to the respective tubes labelled and again vortexed and incubated at 35° C/2 hours. Positive tests showed a change in color (yellow as positive and unchanged as negative). E. coli ATCC 25922 was used as negative control and K. pneumoniae ATCC BAA-1705 as positive control. The interpretation of the test was according to CLSI M100 2023 [19]. And Line 242 However with CARBA NP strain C1996 was interpreted as positive as well as C4256., Previously C4067 was defined to be negative with mCIM however, once repeated with CARBA NP was defined as positive.

- Given the limitations of the current assay, would exploring alternative methods for carbapenemase detection, such as molecular techniques (e.g., PCR), be a valuable addition to strengthen the study's findings?

About this comment, actually is not necesary since our work included whole genoma sequencing and the look and informatiojn about resistome was presented and included in fig and text. However, we decided to prove it with CARBA NP as you recommeded.

- Can you suggest any modifications or optimizations to the current phenotypic assay that might improve its applicability to a broader range of microorganisms?

How do you plan to address the limitations of the current approach in future studies investigating carbapenemase activity in these specific microorganisms?

This can be carried on in future studies, no doubt that changes can be performed. Thank you.

We want to say you thank you for your time and effort.

Reviewer #2:

Thank you very much for your comments.

There are mentions of additional research in the manuscript in line 337 - 338 and 362-363; authors are advised to state them in the conclusion section.

We want to thank you about this observation we added the next in Conclusions “More studies are needed to prove that strains C2723, C1519 and C4067 are new species within the Myroides genus.”

Authors should provide study limitations in the manuscript

Thank you for the advice, we have added this one as “Certainly, our work has limitations, design nature without doubts since is centered only the background of clinical strains of Myroides spp., the number of strains included, in this sense information about prevalence around the world is limited beside we included all references published. The correlation with a in vivo model could evaluate the impact of virulence factors and the need of phenotypic methods to explore the activity of carbapenemases in this genus”

A typo/unclear statement in line 371. Authors are advised to edit this typographical error.

Thank you so much for your kind revision, advice and support, we have changed to The Myroides genus is a rare microorganism that can cause infections in both immunosuppressed and immune competent hosts. The outcome of infection varies depending on the antimicrobial profile of the bacteria. Molecular techniques are essential for accurately identifying these bacteria, characterizing potential species, and exploring therapeutic options for achieving clearance of infections.

Thank you so much for everything.

Reviewer #3:

Lines 137-149. Author should cite a reference for the method used for whole genome sequencing.

Thank you so much for, we looked for how to cite, and in several papers describe the procedure according to manufacturer, however, we include reference, number 15.

Lines 150-156

• Authors should cite a reference for the method used here too.

• Methods should be well cited.

Citation was done. Thank you for the advice.

Lines 371-371

• Myroides genus is not such as uncommon microorganism causing infections, both in

• immunosuppressed and immune competent hosts, with different outcomes depending on

• antimicrobial profiles. WHY NOT SAY: Myroides genus is a common microorganism causing...

Thank you, change was done as following: The Myroides genus is a rare microorganism that can cause infections in both immunosuppressed and immune competent hosts. The outcome of infection varies depending on the antimicrobial profile of the bacteria. Molecular techniques are essential for accurately identifying these bacteria, characterizing potential species, and exploring therapeutic options for achieving clearance of infections.. More studies are needed to prove that strains C2723, C1519 and C4067 are new species within the Myroides genus.

Page 15

• Are there more specific ways to itemize the data availability Statement.

• Ensure regular spacing through the write up.

In conclusion, authors should identify any limitations to the execution of the research as this is necessary and obvious going by the methods used.

About numeration I´ll check it in the page when we generate the PDF, thank you.

About the ways more specific, I really Will appreciate if you help me with this point for future Works, thank you for you support.

I´ll check the spaces.

Thank you for your time, kindness, support and advices. We really appreciate it.

Editor Comments:

Thank you very much for your comments.

1. Lines 120-124: Please provide the range of concentrations for each antimicrobial tested.

Thank you. We have added For AK, GN, ATM, CAZ and FEP concentrations tested were from 256 µg/mL to 0.25 µg/m, while for CIP, LVX, MEM, IMP, AZT, E, DOX, and TYG concentrations were 64 µg/mL to 0.0612 µg/mL. On the other hand for COL and TZP were 128 µg/mL to 0.125 µg/mL and 128/4 µg/mL to 0.125/4 µg/mL respectively. Finally, SXT starting concentration 8/152 µg/mL to 0.00078/0.1484 µg/mL.

Lines 362–368: This is not clear. Please rewrite for clarity and flow.

Thanks a lot for your advice. Change was made according recommendation as follow: Pathogen Myroides species have been shown to possess a number of VF that are associated with a range of processes, including cell adherence, intracellular survival, and capsule production. Nevertheless, a more comprehensive investigation into the molecular mechanisms underlying the pathogenicity of Myroides spp. has yet to be conducted. Therefore, the current state of knowledge regarding the infection process and disease-causing mechanisms of Myroides is limited. Based on the identified VF, the results suggest that the isolates under study possess the ability to survive and replicate within host cells, evade phagocytic cells of the immune system, and are able to cause disease. This is a virulence trait observed in other opportunistic pathogens, including Mycobacterium spp., Pseudomonas spp., Serratia spp., and others

Highlight the key limitations of the present study in the last paragraph of the discussion section.

Thank you, limitations were added.

Certainly, our work has limitations, design nature without doubts since is centered only the background of clinical strains of Myroides spp., the number of strains included, in this sense information about prevalence around the world is limited beside we included all references published. The correlation with a in vivo model could evaluate the impact of virulence factors and the need of phenotypic methods to explore the activity of carbapenemases in this genus.

In your conclusion, can you provide some recommendations based on your study findings?

Thank you, we added the next “Finally, the species analyzed in the present work showed reduced susceptibility patterns for aminoglycosides, ß-lactams, cyclic lipopeptides, this was corroborated with the genes found in the genome, therefore, susceptibility tests should be performed in the clinical microbiology laboratory and the Infectious diseases specialists staff should consider these susceptibility patterns for the treatment scheme of patients suffering infections by microorganisms of this genus”.

We would like to thank all the reviewers for their time, kindness and advice, which will enrich our work. Thank you very much for your time, kindness and advice, which will allow us to enrich our work.

---

## [Decision Letter · Decision Letter 1]

28 Aug 2024

Myroides species, pathogenic spectrum and clinical microbiology sight in Mexican isolates.

PONE-D-24-22192R1

Dear Dr. Jacome,

We’re pleased to inform you that your manuscript has been judged scientifically suitable for publication and will be formally accepted for publication once it meets all outstanding technical requirements.

Kind regards,

Mabel Kamweli Aworh, DVM, MPH, PhD. FCVSN

Academic Editor

PLOS ONE

Additional Editor Comments (optional):

Reviewers' comments:

Reviewer's Responses to Questions

**Comments to the Author**

1. If the authors have adequately addressed your comments raised in a previous round of review and you feel that this manuscript is now acceptable for publication, you may indicate that here to bypass the “Comments to the Author” section, enter your conflict of interest statement in the “Confidential to Editor” section, and submit your "Accept" recommendation.

Reviewer #1: All comments have been addressed

Reviewer #2: All comments have been addressed

2. Is the manuscript technically sound, and do the data support the conclusions?

Reviewer #1: Yes

Reviewer #2: Yes

3. Has the statistical analysis been performed appropriately and rigorously? 

Reviewer #1: N/A

Reviewer #2: Yes

4. Have the authors made all data underlying the findings in their manuscript fully available?

Reviewer #1: Yes

Reviewer #2: Yes

5. Is the manuscript presented in an intelligible fashion and written in standard English?

Reviewer #1: Yes

Reviewer #2: Yes

6. Review Comments to the Author

Reviewer #1: Authors have addressed all my concerns and i do not have any other concerns with this submission. Great Job

Reviewer #2: The authors have responded appropriately to the suggested recommendations and corrections have been effected. The manuscript is technically sound and presented in an intelligible fashion.

7. PLOS authors have the option to publish the peer review history of their article (what does this mean?). If published, this will include your full peer review and any attached files.

Reviewer #1: **Yes: **Abdulhakeem Abayomi Olorukooba

Reviewer #2: **Yes: **Olubunmi Margaret Ogbodu

---

## [Editor Report · Acceptance letter]

2 Sep 2024

PONE-D-24-22192R1 

PLOS ONE

Dear Dr. López-Jácome, 

I'm pleased to inform you that your manuscript has been deemed suitable for publication in PLOS ONE. Congratulations! Your manuscript is now being handed over to our production team.

Kind regards, 

on behalf of

Dr. Mabel Kamweli Aworh 

Academic Editor

PLOS ONE